# The hammam effect or how a warm ocean enhances large scale atmospheric predictability

Davide Faranda[1,2], M. Carmen Alvarez-Castro [1,3], Gabriele Messori[1,4,5], David Rodrigues[1] & Pascal Yiou[1]

The atmosphere's chaotic nature limits its short-term predictability. Furthermore, there is little knowledge on how the difficulty of forecasting weather may be affected by anthropogenic climate change. Here, we address this question by employing metrics issued from dynamical systems theory to describe the atmospheric circulation and infer the dynamical properties of the climate system. Specifically, we evaluate the changes in the sub-seasonal predictability of the large-scale atmospheric circulation over the North Atlantic for the historical period and under anthropogenic forcing, using centennial reanalyses and CMIP5 simulations. For the future period, most datasets point to an increase in the atmosphere's predictability. AMIP simulations with $4K$ warmer oceans and $4 \times$ atmospheric $CO_2$ concentrations highlight the prominent role of a warmer ocean in driving this increase. We term this the hammam effect. Such effect is linked to enhanced zonal atmospheric patterns, which are more predictable than meridional configurations.

[1] Laboratoire des Sciences du Climat et de l'Environnement LSCE-IPSL, CEA Saclay l'Orme des Merisiers, UMR 8212 CEA-CNRS-UVSQ, Université Paris-Saclay, 91191 Gif-sur-Yvette, France. [2] London Mathematical Laboratory, 8 Margravine Gardens, London W68RH, UK. [3] Climate Simulation and Prediction Division, Centro Euro-Mediterraneo sui Cambiamenti Climatici, Bologna 40127, Italy. [4] Department of Earth Sciences, Uppsala University, Uppsala 75236, Sweden. [5] Department of Meteorology, Stockholm University and Bolin Centre for Climate Research, Stockholm 10691, Sweden. Correspondence and requests for materials should be addressed to D.F. (email: davide.faranda@cea.fr)

Will the difficulty of forecasting weather be affected by climate change? To answer this question, the first step is to recognise the chaotic nature of atmospheric dynamics[1]. Despite the increase in resolution and complexity of weather forecast systems, atmospheric forecasts face an insurmountable predictability limit[2]. This stems from the intrinsic properties of the atmospheric attractor—a high-dimensional geometric object on which all the possible atmospheric states settle—and is often referred to as the "butterfly effect", or dependence on initial conditions[3]. Atmospheric turbulence injects energy at all spatial and temporal scales, generating chaos and limiting short-term predictability. Predictability further depends on space and time, so that the detection and interpretation of changes is extremely complex[4]. Indeed, predictability is not a constant, and can be strongly affected by the large-scale atmospheric configuration from which the forecast is initialised[5]. For example, mid-latitude transitions from blocked to zonal atmospheric states are less predictable than a persistent large-scale zonal flow[6]. By large-scale zonal flow, we refer to a strong eastward (zonal) jet with few meridional oscillations. If the proportion of predictable vs. unpredictable patterns were to change under anthropogenic forcing, the atmosphere's intrinsic predictability would also change accordingly. The task of diagnosing changes in atmospheric predictability—and more generally in mid-latitude atmospheric dynamics—under anthropogenic forcing is therefore beset with difficulties[7–9]. Achieving this when focusing on averaged quantities, such as climate variability indices, presents serious challenges[10].

Here, we diagnose predictability by relying on two objective metrics that sample the properties of the atmospheric attractor. Such metrics provide information on the number of active degrees of freedom of the system and on the typical timescales of the flow, and hence a measure of the intrinsic predictability of an atmospheric state. This is in contrast to predictability defined relative to the performance of a numerical weather prediction model. Computing such metrics was a major challenge until recently[11,12]. However, advances in dynamical systems theory now allow us to compute them for instantaneous atmospheric states[13]. Their calculation is based on analogues, or recurrences, of instantaneous (weather) patterns. Recent results have demonstrated their effectiveness in classifying weather patterns leading to climate extremes in the North Atlantic region[13,14], as well as hemisphere-wide atmospheric variability[15,16].

We specifically focus on evaluating changes in the intrinsic predictability of the atmospheric circulation over the North Atlantic under anthropogenic forcing, as represented by the above dynamical systems indicators. In the historical period, there is disagreement between reanalysis datasets. For the future period, most of the projections analysed here point to an increase in the atmosphere's intrinsic predictability. The analysis of Atmospheric Model Intercomparison Project (AMIP) simulations with 4 K warmer oceans and four times atmospheric $CO_2$ concentrations points to the prominent role of a warmer ocean in driving this increase. We name this robust signal the hammam effect. This occurs through the enhancement of zonal atmospheric patterns, which are more predictable than meridional configurations.

## Results

### Data and predictability metrics

Our analysis is based on two dynamical systems metrics: the local (in phase space) dimension $d$ and the persistence $\theta^{-1}$ [17]. In simple terms, they describe the recurrences of a system around a state $\zeta$ in phase space. In our case, $\zeta$ would be a latitude–longitude map of a given variable for a given day and dataset. Values of $d$ and $\theta^{-1}$ are obtained for every time-step (i.e. every state $\zeta$) in the dataset of interest. $d$ provides information on how the system can reach $\zeta$ and how it can evolve from $\zeta$, and is a proxy for the system's active number of degrees of freedom. This information is intrinsically linked to the predictability of $\zeta$, but is local in nature. $d$ can change rapidly over timescales of a few days meaning that, when applied to atmospheric fields, it cannot be interpreted in the same way as the error of a medium-range weather forecast initialised from $\zeta$. $\theta^{-1}$ describes the persistence of $\zeta$ in time. A very persistent state is typically highly predictable, while a very unstable state yields low persistence. In this sense, the information provided by $\theta^{-1}$ is more directly linked to that provided by the error of a forecast initialised from $\zeta$[18]. $d$ and $\theta^{-1}$ therefore provide complementary information on the intrinsic predictability of an instantaneous state of the atmosphere. Details of how these two metrics are computed are provided in the Methods. We compute the above two dynamical systems metrics for the sea-level pressure field in the North Atlantic region (22.5°N–70°N and 80°W–50°E) for several datasets. These include the three longest reanalysis ensembles available over our domain, namely: the National Ocean and Atmosphere Administration's (NOAA) atmospheric reanalysis of the twentieth century (20CRv2c)[19] with 56 members covering the period 1851–2014; the European Centre for Medium Range Weather Forecasts' (ECMWF) atmospheric model integrations of the twentieth century (ERA20CM, as ERA20C but with no synoptic meteorological data assimilated and all observational information incorporated in the boundary conditions and forcing[20]), with 10 members covering the period 1900–2100; and ECMWF's coupled climate reanalysis of the twentieth century (CERA20C)[21] with 10 members covering the period 1900–2010. They are complemented by a suite of state-of-the-art climate model simulations, including Coupled Model Intercomparison Project phase 5 (CMIP5) model simulations from 1850 to 2100 (see Methods and Supplementary Table 1 for the exact periods used in this study). The choice of region is motivated by the greater abundance of observations in the North Atlantic during the historical period than in other parts of the globe. The choice of SLP is similarly motivated by the fact that it is well constrained in the long reanalyses, as well as being representative of the main large-scale variability modes and atmospheric features of the North Atlantic region[22]. The versatility of our metrics allows for their use in transient simulations with different horizontal resolutions, without the need for regridding or detrending. In order to corroborate our findings, we further analyse AMIP simulations with atmospheric $CO_2$ concentrations increased by a factor four and ocean temperatures increased by 4 K. In ref. [13] and in this study $d_{SLP}$ ~12–13 for the North Atlantic. This is consistent with the results of ref. [23] that demonstrate that the low-frequency variability in the Northern Hemisphere is spanned by around half-a-score empirical normal modes, with growth rates of around 3–5 days in phase space.

We also compute $d$ for the SST fields of some of the above datasets. Hereafter, we specify whether we are referring to $d_{SLP}$ or to $d_{SST}$. We do not adopt this notation for $\theta$ as we compute it for the SLP field only. Finally, we note that the multi-model and ensemble-average $d$ and $\theta$ discussed below are not the metrics computed on the mean SLP fields, but rather the mean values of the metrics computed on the SLP fields of each model or ensemble member. This avoids introducing spurious trends in the data[16]. Since we are interested in long-term changes in predictability, we apply a 5-year running average to each quantity.

### Observed and projected predictability changes

We first comment on the historical period and the reanalysis datasets (Fig. 1 and Table 1). The absolute values of $d_{SLP}$ and $\theta$ for all datasets are

reported in Supplementary Fig. 1. The 20CRv2c ensemble shows a significant decreasing trend in $d_{SLP}$, whereas the inverse persistence increases only over the early period of the reanalysis, before stabilising. This is possibly an effect of the scarcity of observational data in the 19th century and the consequent large spread of the members[24]. The ERA20CM and CERA20C reanalyses broadly agree with 20CRv2c in $\theta$, as they show a weak neutral and positive trend, respectively, but only cover the period where the 20CRv2c $\theta$ stabilises. Important discrepancies emerge in $d_{SLP}$: ERA20CM displays no significant trend, while CERA20C displays a significant increasing trend, opposite in sign to the significant decreasing trend seen in 20CRv2c. All three datasets also display large interdecadal fluctuations, with ERA20CM showing a lower variability than the other two. It is encouraging to note that, during the common period, CERA20C shows the same interdecadal variability as 20CRv2c; nonetheless, the discrepancy in the long-term $d_{SLP}$ trends between the three reanalyses remains to be explained. The main difference between ERA20CM and CERA20C is the presence, in the latter, of data assimilation and a coupled ocean[21]. To save computational time, it is common practice to restart the assimilation process of the reanalyses every 10 years[25]. We speculate that this operation may affect the ocean dynamics because it does not give them time to settle on the attractor, as the trajectory is displaced every 10 years. On the contrary, it affects only marginally the atmosphere-only reanalyses without an active ocean component. This does not, however, explain the differences between ERA20CM and 20CRv2c, which must derive from differences in the numerical models and data assimilation schemes.

The historical CMIP5 simulations (Fig. 1 and Table 1) display a significant decreasing trend in $d_{SLP}$, whereas $\theta$ shows an increasing, albeit very weak, trend. Unlike the reanalyses, CMIP5 data does not display large interdecadal fluctuations in the two dynamical quantities. This is partly due to the fact that, since we

are considering a multi-model mean, the interdecadal fluctuations between individual members average out. However, even when individual historical model runs are analysed, the standard deviations of $d_{SLP}$ and $\theta$ are smaller than in 20CRv2c and CERA20C, and comparable to those of the ERA20CM dataset. These values increase to levels comparable to—or even exceeding—those of the two former datasets in the representative concentration pathway (RCP) scenarios.

We next discuss future climates using CMIP5 simulations run under the RCP 4.5 and 8.5 scenarios, as well as the Community

**Table 1 Mann–Kendall test p values for $d_{SLP}$, $\theta$ and the 5th and 95th percentiles of $d_{SLP}$ at the 5% significance level**

| Ensemble | p Value | | | | Trend | | | |
|---|---|---|---|---|---|---|---|---|
| | d | θ | 5th | 95th | d | θ | 5th | 95th |
| 20CRv2c | 7.5e−05 | 2.6e−05 | 2.4e−4 | 0.01 | ⇓ | ⇑ | ⇓ | ⇓ |
| ERA20CM | 0.87 | 0.92 | 0.53 | 0.50 | = | = | = | = |
| CERA20C | 0.0297 | 0.001 | 0.38 | 0.01 | ⇑ | ⇑ | = | ⇑ |
| CESM | 1.1e−05 | 0.23 | 0.35 | 0.013 | ⇓ | = | = | ⇓ |
| CMIP5 Hist | 4.7e−05 | 3.1e−04 | 0.08 | 0.004 | ⇓ | ⇑ | = | ⇓ |
| CMIP5 RCP 4.5 | 3.9e−06 | 0.83 | 1.5e−4 | 3.8e−6 | ⇓ | = | ⇓ | ⇓ |
| CMIP5 RCP 8.5 | 1.05e−07 | 0.04 | 5.4e−6 | 6.7e−5 | ⇓ | ⇑ | ⇓ | ⇓ |

Arrows (⇓/⇑) represent significant decreasing/increasing trends at the 5% level, respectively. The equal sign (=) denotes absence of significant trends. The test statistics is described in Supplementary Note 1
*SLP* sea-level pressure, *CESM* Community Earth System Model, *CMIP5* Coupled Model Intercomparison Project phase 5, *RCP* representative concentration pathway, *CERA20C* ECMWF's coupled climate reanalysis of the twentieth century, *ERA20CM* ECMWF's atmospheric model integrations of the twentieth century, *20CRv2c* NOAA's 20th century atmospheric reanalysis version 2c

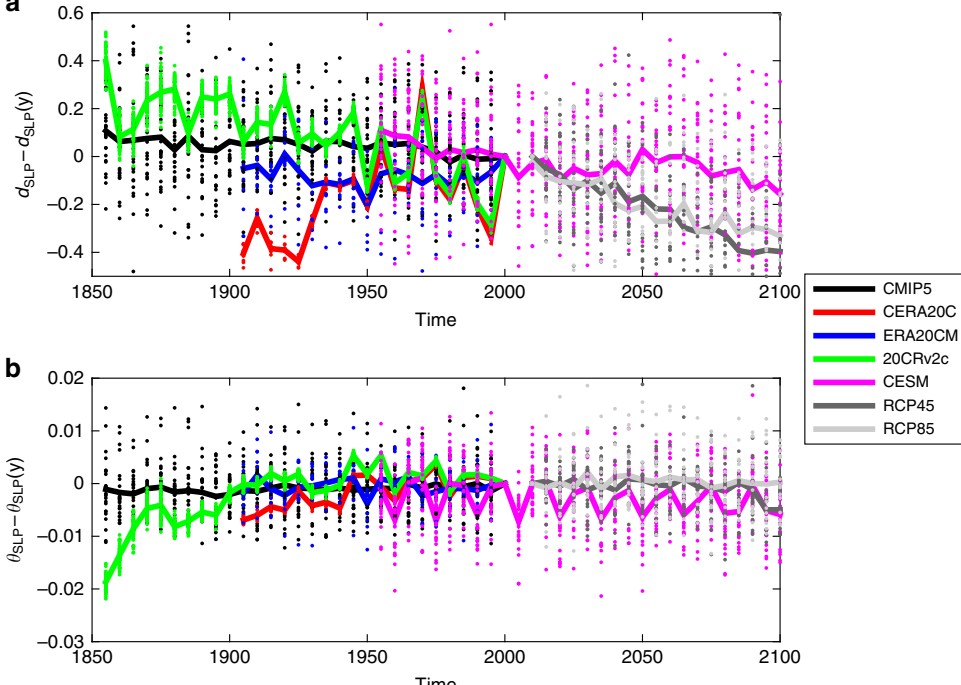

**Fig. 1** Local dimension and inverse persistence for all the datasets. Five-year averages of local dimension $d_{SLP}$ (**a**) and inverse persistence $\theta_{SLP}$ (**b**) minus the respective values $d_{SLP}(y)$ and $\theta_{SLP}(y)$ computed for the years $y = 2000$ (or $y = 2006$ for representative concentration pathway (RCP) scenarios). Different colours correspond to different datasets as shown in the legend. Dots: single members or models. Solid lines: means of the ensembles. SLP, sea-level pressure

Earth System Model (CESM) RCP 8.5 large ensemble. The CMIP5 models show a significant decreasing trend in $d_{SLP}$ and a weak increasing trend in $\theta$ for the RCP 8.5 scenario (Fig. 1 and Table 1). CESM shows similar results for $d_{SLP}$ but no significant change in $\theta$. There is a large spread among the CESM members—larger in fact than that among the different CMIP5 models. We also note that there is no large difference between the RCP 4.5 and RCP 8.5 $d_{SLP}$ trends. With the exception of the two ECMWF reanalyses, all datasets therefore display a decrease in $d_{SLP}$ throughout the period 1850–2100, albeit in some cases modulated by a marked inter-decadal variability. This trend is primarily associated with the summer (June–August, JJA) and autumn (September–November, SON) seasons (see Supplementary Fig. 2 and Supplementary Tables 2 and 3). Overall, the datasets with decreasing trends show a relative variation of $d_{SLP}$ of about 5–8%.

In order to understand the origin of this trend, we revert to the analysis of the atmospheric configurations prevailing in each dataset. In general, high $d$ are associated with low-predictability meridional configurations, such as blocking or mid-Atlantic ridges. Low $d$ match instead high-predictability zonal configurations[13,14]. A shift of the upper and lower percentiles of the $d_{SLP}$ distributions would therefore correspond to an enhanced zonality or meridionality of the flow. To test this idea, we repeat the analysis shown in Fig. 1 for the top and bottom five percentiles of the $d_{SLP}$ distributions (see Supplementary Fig. 3). We find that the trends are consistent with those found for the mean values, implying an enhanced zonality of the flow associated with a decrease in mean $d_{SLP}$. These results are also consistent with the dominant role of the summer and autumn seasons in the $d$ trends. Indeed, the summer NAO displays a long-term trend towards more positive values in the future, corresponding to an enhanced zonality of the flow[26]. A future

year-round increase in flow zonality over the North Atlantic, peaking in autumn, was also found by de Vries et al. [27]. The dynamical significance of our results is illustrated by Faranda et al.[28], where a low-dimensional model of the jet dynamics over the northern hemisphere is derived. The authors found that the jet dynamics is sensitive to small shifts in a control parameter (the meridional temperature gradient) and that the dimension $d$ is proportional to the number of discontinuities of the jet (a proxy of blocking). Finally, the observed jet dynamics correspond to a region of the parameter space that is sensitive to perturbations, where small changes in the dimension correspond to large dynamical changes. In the dynamical system jargon, this means that the system is close to bifurcation points.

**The hammam effect**. What could be the root driver of the above changes? To investigate this, we analyse two sets of forced AMIP simulations: the first with $4 \times CO_2$ and the second with a 4 K warmer ocean. In the former, energy is injected immediately everywhere into the atmosphere, whereas in the latter the energy is stored in the ocean and may then affect the atmosphere through surface interactions. In the $4 \times CO_2$ runs, there is no significant change in $d_{SLP}$ or $\theta$ (Fig. 2). Therefore, the greenhouse enhancement does not appear to radically alter the dynamical properties of atmospheric motions. There is instead a significant change in the dynamical properties when the ocean is 4 K warmer. In this world, which one may imagine akin to a Turkish bath, or hammam, the local dimension decreases (Fig. 2) and the inverse persistence increases (not shown). An intuitive explanation for this phenomenon could be that a warmer ocean implies a stronger mid-latitude jet and the partial suppression of meridional patterns, such as blocking or Atlantic Ridges[29]. This leads

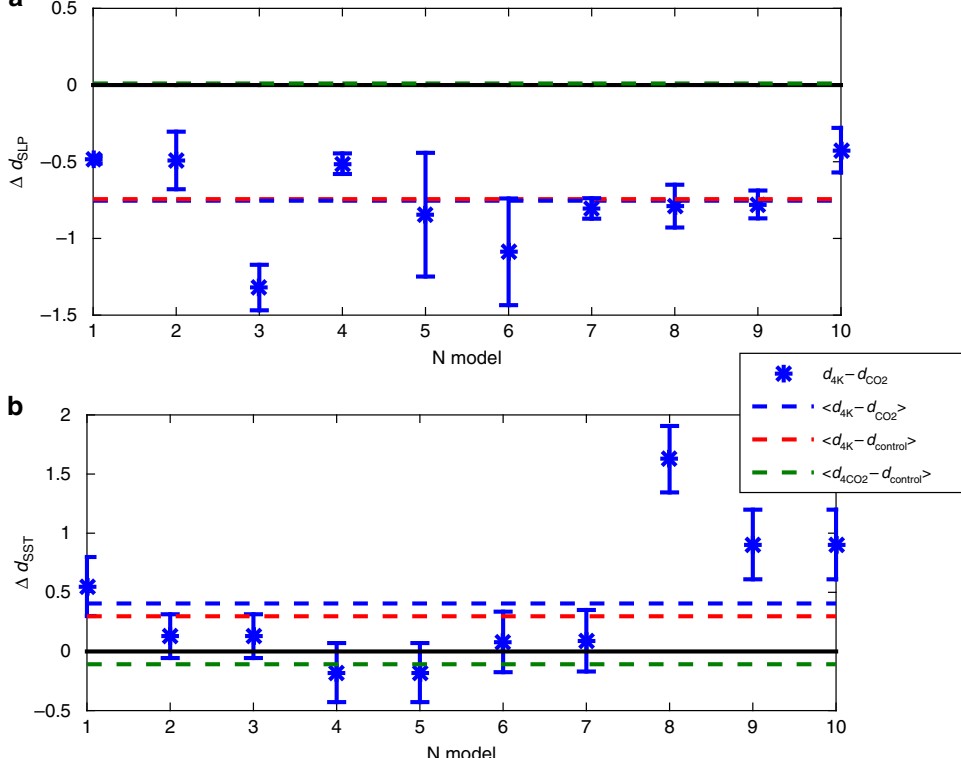

**Fig. 2** Differences of local dimension in $4 \times CO_2$ and $+4$ K Atmospheric Model Intercomparison Project (AMIP) simulations. Differences of $\Delta d$ between average local dimension $d_{SLP}$ for daily sea-level pressure (SLP) data (**a**, **b**) and $d_{SST}$ for monthly sea-surface temperature (SST) fields for the $4 \times CO_2$ and $+4$ K AMIP simulations with respect to the control runs. Error bars indicate the standard deviation of the mean. Lines: means of the ensembles, indicated in the legend by angular brackets

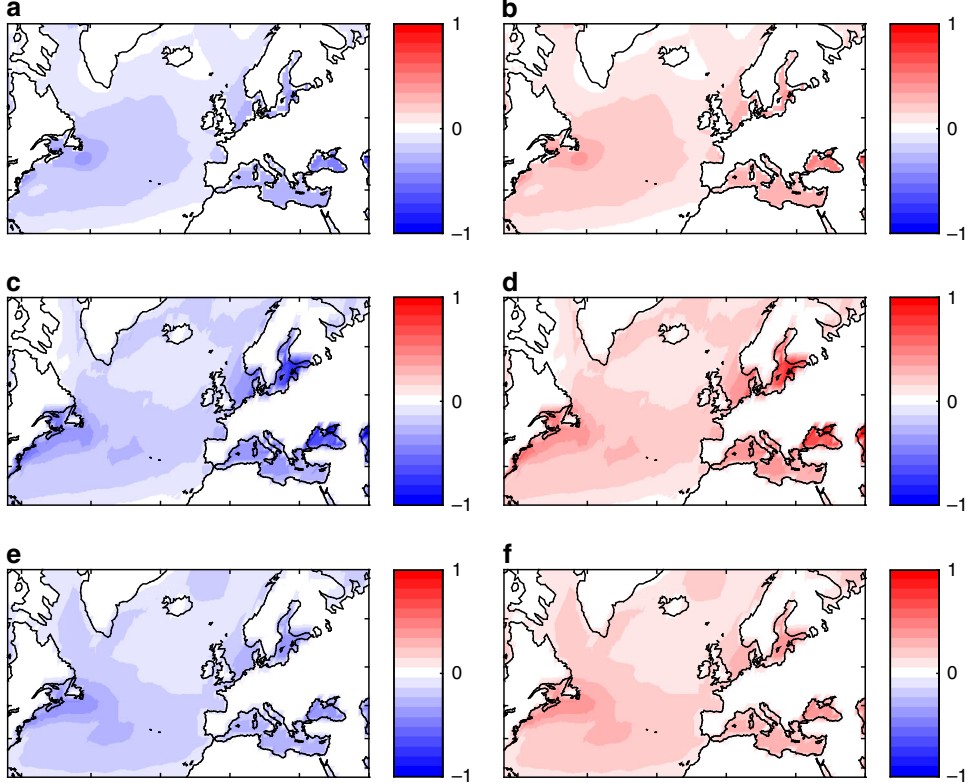

**Fig. 3** Composite sea-surface temperature anomalies for high and low local dimension. Composite sea-surface temperature anomalies (units: K) with respect to the climatological seasonal cycles for days with $d_{SST}$ above (**a**, **c**, **e**) and below (**b**, **d**, **f**) the respective median values. Datasets: COBE, as used in 20CRv2c (**a**, **b**), ERA20CM (**c**, **d**) and CERA20C (**e**, **f**)

us to hypothesise that the long-term trends in $d_{SLP}$ discussed above may be associated to changes in SSTs, with higher SSTs in the North Atlantic corresponding to lower local dimensions in the atmosphere. We confirm this inference by computing the local dimension of the SSTs from the subset of CMIP5 models that have both pre-industrial control and 4 K warmer ocean runs. We find a significant increase in the average $d_{SST}$ across the warmer simulations, pointing to concomitant and opposite in sign variations in the atmospheric and oceanic local dimensions, at least in the models analysed here (Fig. 2b).

We next verify whether a clear dependence between SST values and $d_{SST}$ is also found in the reanalysis datasets. Above-median values of $d_{SST}$ correspond to widespread cold anomalies across the Atlantic basin, while below-median values display anomalies of the opposite sign (Fig. 3). This is not directly comparable to Fig. 2b, as we are looking at SST anomalies here. Nonetheless, it highlights a close coupling between $d_{SST}$ and basin-wide SSTs. For the historical period, all three reanalysis datasets display rapidly increasing SSTs (not shown), yet only one of these datasets (20CRv2c) displays a significant decrease in $d_{SLP}$, as seen in the models. This points to differences in the ocean–atmosphere coupling as the key to the discrepancies between the three reanalyses. The CMIP5 simulations are coherent with the coupling found in the 20CRv2c reanalysis, since they display a decreasing $d_{SLP}$ trend and a gradual increase in SSTs from the beginning of the historical period to the end of the century[30].

## Discussion

We have shown that the atmospheric circulation's intrinsic predictability in the North Atlantic has increased in the recent past, and will continue to do so in a future with continued high levels

of anthropogenic emissions. The local dimension $d_{SLP}$, which we take to be representative of large-scale atmospheric motions, decreases. At the same time, the persistence $\theta^{-1}$ shows a weaker decrease. Thus, the effect of increasing dimension dominates. The main driver of the trend in $d_{SLP}$ appears to be the warmer (Atlantic) SSTs, which act to strengthen the zonality of the large-scale atmospheric flow and reduce the meridional patterns responsible for the more unpredictable situations. Warmer SSTs also correspond to more predictable atmospheric configurations, since $d_{SLP}$ is lower for warmer ocean states. We named this robust response the hammam effect.

It remains to explain why an increase in the temperature of the climate system, which corresponds to an increase in the Reynolds number for turbulent flows, could lead to more predictable states. For most of the flows, increasing the Reynolds number corresponds to an increase of the number of degrees of freedom. However, there are several examples of the opposite behaviour. A first example is the von Karman flow dynamics: Faranda et al.[31] showed that there is a region of very high Reynolds number where a low-dimensional dynamics emerges out of featureless turbulence. Another example is provided by Faranda et al.[28] who analysed the jet stream dynamics using a low-dimensional model obtained by embedding the data. The bifurcation sequence, when increasing the control parameter (a surrogate of the Reynolds number), evolves from a high-dimensional noisy fixed point to a lower dimensional structure where the dynamics of the jet switches between blocked and zonal flows. Possible explanations of these phenomena of noise-induced order are investigated in the framework of stochastic dynamical systems[32,33].

Our conclusions come with some caveats. First, the ECMWF reanalyses do not show a decrease in dimensionality, but rather

show either an increase of $d_{SLP}$ with time (CERA20C) or no trend (ERA20CM). Although the ERA20CM trend is compatible with those observed in some of the CMIP5 models, the CERA20C case is unique amongst the analysed data. We hypothesise that this may be due to the way the observations are assimilated into the reanalyses. For example, they are reinitialised every ten years, possibly leaving the ocean in an unstable transient state. Differences in ocean–atmosphere coupling between the reanalysis datasets may also play an important role. Moreover, our analysis does not allow us to draw conclusions concerning the global circulation nor small-scale phenomena. Indeed, it relates more to the predictability of large-scale motions than to conventional weather predictability, although a relevant correlation between the variation of local dimensions and persistence and the spread in numerical forecasts of SLP fields exists[13,18]. Finally, the validity of our results is restricted to the North Atlantic region, chosen because of the comparative abundance of observational data during the early reanalysis period. This is, however, also a strength of our methodology: the possibility of focusing on a specific region and scale by choosing the appropriate observable fields allows for future targeted studies addressing changes in predictability occurring, for example, in tropical or monsoonal regions, as well as analyses of the role of small-scale phenomena.

## Methods

**Climate models employed for the analyses.** We analyse daily output of the CMIP5[34] for: 27 historical simulations (Supplementary Table 1), 18 RCP4.5/8.5 projections and ten +4 K sea-surface temperature (SST) and $4 \times CO_2$ atmosphere-only (AMIP) simulations. The historical simulations cover the period 1850–2000; the forcings are consistent with observations and include changes in: atmospheric composition due to anthropogenic and volcanic influences, solar forcing, emissions or concentrations of short-lived species and natural and anthropogenic aerosols or their precursors, as well as land use. RCP4.5 and RCP8.5 projections are projections of future climates (2006–2100) forced by two representative concentration pathway (RCP) scenarios. These result in a radiative forcing of 4.5 and 8.5 W m$^{-2}$, respectively, in year 2100, relative to pre-industrial conditions. AMIP simulations cover the 1979–2005 period and are performed by prescribing SSTs and sea ice boundary conditions to an atmosphere-only model. +4 K simulations impose a uniform 4 K warming to observed SSTs, while the $4 \times CO_2$ simulations impose quadrupled atmospheric $CO_2$ concentrations relative to the pre-industrial control value of 280 ppm. We additionally analyse a 32-member ensemble of simulations from the CESM[35]. This covers the 1950–2100 period, and follows an RCP8.5 scenario starting from 2006. As the historical part of the simulations is much shorter than for the other datasets, we analyse $d$ and $\theta$ trends over the whole CESM period as opposed to performing separate analyses for historical and future trends.

**Dynamical systems notions.** The attractor of a dynamical system is a geometric object defined in the space hosting all the possible states of the system (phase-space). Each point $\zeta$ on the attractor can be characterised by two dynamical indicators: the local dimension ($d$), which indicates the number of degrees of freedom active locally around $\zeta$, and the persistence ($\theta^{-1}$), a measure of the mean residence time of the system around $\zeta$[13].

**Local dimension and persistence.** To determine $d$, we exploit recent results from the application of extreme value theory to Poincaré recurrences in dynamical systems. This approach considers long trajectories of a system—in our case successions of daily SLP latitude–longitude maps—corresponding to a sequence of states on the attractor. For a given point $\zeta$ in phase space (e.g. a given SLP map), we compute the probability that the system returns within a ball of radius $\epsilon$ centred on the point $\zeta$. The Freitas et al.[36] theorem, modified by Lucarini et al.[37], states that logarithmic returns:

$$g(x(t)) = -\log(\text{dist}(x(t), \zeta)) \tag{1}$$

yield a probability distribution such that:

$$\Pr(z > s(q)) \simeq \exp\left[ -\vartheta(\zeta)\left( \frac{z - \mu(\zeta)}{\sigma(\zeta)} \right) \right], \tag{2}$$

where $z = g(x(t))$ and $s$ is a high threshold associated to a quantile $q$ of the series $g(x(t))$. Requiring that the orbit falls within a ball of radius $\epsilon$ around the point $\zeta$ is equivalent to asking that the series $g(x(t))$ is over the threshold $s$; therefore, the ball radius $\epsilon$ is simply $e^{-s(q)}$. The resulting distribution is the exponential member of the Generalised Pareto Distribution family. The parameters $\mu$ and $\sigma$, namely the location and the scale parameter of the distribution, depend on the point $\zeta$ in phase space. $\mu(\zeta)$ corresponds to the threshold $s(q)$, while the local dimension $d(\zeta)$ can be obtained via the relation $\sigma = 1/d(\zeta)$.

When $x(t)$ contains all the variables of the system, the estimation of $d$ based on extreme value theory has a number of advantages over traditional methods (e.g. the box counting algorithm[38,39]). First, it does not require to estimate the volume of different sets in scale space: the selection of $s(q)$ based on the quantile provides a selection of different scales $s$, which depends on the recurrence rate around the point $\zeta$. Moreover, it does not require the a priori selection of the maximum embedding dimension as the observable $g$ is always a univariate time-series.

The persistence of the state $\zeta$ is measured via the extremal index $0 < \vartheta(\zeta) < 1$, an adimensional parameter, from which we extract $\theta(\zeta) = \vartheta(\zeta)/\Delta t$. $\theta(\zeta)$ is therefore the inverse of the average residence time of trajectories around $\zeta$ and it has unit of a frequency (in this study 1/days). If $\zeta$ is a fixed point of the attractor $\theta(\zeta) = 0$. For a trajectory that leaves the neighbourhood of $\zeta$ at the next time iteration, $\theta = 1$. To estimate $\theta$, we adopt the Süveges estimator[40]. For further details on the extremal index, see ref. [17].

## Code availability

The code for the computation of the local dimension is available from https://www.lsce.ipsl.fr/Pisp/davide.faranda/ and can also be requested by email from the corresponding author.

## Data availability

The datasets analysed in this paper are available in the 20CRv2c repository: https://www.esrl.noaa.gov/psd/thredds/catalog/Datasets/20thC_ReanV2c /Dailies/gaussian/monolevel/catalog.html; in the COBE-SST2 repository: https://www.esrl.noaa.gov/psd/thredds/catalog/Datasets/COBE/catalog.html; in the ERA20CM repository: https://apps.ecmwf.int/datasets/data/era20cm/; in the CERA20C repository: https://apps.ecmwf.int/datasets/data/cera20c/; in the CMIP5 repository: https://esgf-node.llnl.gov/projects/cmip5/; and in the Large Ensemble Community Project repository: http://www.cesm.ucar.edu/projects/community-projects/LENS/data-sets.html.

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

## Acknowledgements

We thank A. Jézéquel, M. Vrac, F. Daviaud, Y. Sato, V. Lucarini and B. Dubrulle for useful discussions. We acknowledge G. Brunet and A. Pouquet for their constructive comments. We also thank J.Y. Peterschmitt for the support in downloading the data. D.F., M.C.A.-C., G.M. and P.Y. acknowledge the support of the ERC Grant No. 338965-A2C2. M.C.A.-C. was further supported by Swedish Research Council Grant No. C0629701 and G.M. was supported by a grant from the Department of Meteorology of Stockholm University and by the Swedish Research Council Grant No. 2016-03724.

## Author contributions

D.F., M.C.A.-C. and G.M. downloaded and processed the data. D.F. carried out the analysis. All the authors co-designed the study and contributed to writing and editing the manuscript.

## Additional information

**Competing interests:** The authors declare no competing interests.

