## [Peer review file · Nature Communications]

Reviewer #1 (Remarks to the Author):

The hammam effect: how a warm ocean makes weather forecasting easier

By D. Faranda, M.C. Alvarez-Castro, G. Messori, D. Rodrigues and P. Yiou

General comments

I recommend that this paper be published, if my main concerns are properly addressed.

This study addresses the evolution of North Atlantic weather predictability within the framework of climate change. It claims using centennial reanalysis datasets and CMIP5 simulations (1851-2100 period) that weather predictability for the North Atlantic region should increase. The main explanation provided for this positive trend is warmer North Atlantic SSTs that would make more zonal the atmospheric circulation hence more predictable.

The methodology of the diagnostics is relatively novel and promising (e.g., Faranda et al., Scientific reports, 2017). Overall I agree that the phase-space framework combined with proper statistical analyses is a capable approach to tackle this complex problem of predictability and dynamical processes.

I have no objection with the statistical methodology used in this work. My main concerns are the dynamical and physical interpretations of the findings and some unclear remarks.

1) What are the explicit values in 2000 for inverse persistence and local dimension in Table 1 and Figure 1? It is very difficult to assess the importance of these results with not having the absolute numbers. As an example in Figure 1a we see that the trends span at most one degree of freedom, but relatively to what? In Faranda et al. (Scientific reports, 2017) they provide a local dimension of around 12-13. This is consistent with the results of Brunet (JAS, 1994) that demonstrate that the low frequency variability (subseasonal to seasonal, S2S, time scale) is spanned by around 8 empirical normal modes in the North Hemisphere with growth rate ranging 3-5 days in phase space. The proposed methodology in this study is not able to capture the small-scale phenomena as mention rightly in Faranda et al. (Scientific reports, 2017), but the numerical weather prediction problem lies in great part in that variability (e.g., transient storms). In Brunet (JAS, 1994) small-scale phenomena project mostly into the phase speed continuous spectrum which is associated with transient atmospheric phenomena.

So this study is about S2S predictability (e.g., weather regimes), not about weather predictability.

In addition if we assume that the S2S component of the local attractor as a dimension of the order $O(10)$ as discussed in Faranda et al. (Scientific reports, 2017) and Brunet (1994) then we need to

explain convincingly here why a small reduction of the dimension of the local attractor by less than one degree of freedom seems to have relatively an important impact on S2S predictability and the circulation configuration (e.g., more flow zonality). Are these two results simply fortuitous or wrong?

2) The study at many occasions tries to explain the discrepancy of results between different reanalyses and modelling systems. The discussion is not convincing. Are these reanalyses and numerical simulations up to the task to detect such trends in the dimension of the local attractor and in the persistence? In Shepherd (Nature Geoscience 7, 2017) the case is clearly made that dynamical processes, like regional circulation over the North Atlantic, are prone to significant inconsistency between different climate models due to model error, especially with precipitation processes.

3) I don't understand the purpose of this discussion about SST local dimension. Why an increase in SST local dimension should be consistent with a decrease in the atmosphere local dimension?

4) P. 9 "The warmer SSTs also induce more predictable boundary conditions for atmospheric motions." Why?

5) The title is misleading. As mentioned above this study is not about weather prediction (i.e. small scale phenomena like storms), but prediction of the large scale atmospheric features of the North Atlantic (e.g., blockings) which of course indirectly modulate in part the weather predictability. In addition the study doesn't quantify the predictability trend in absolute numbers, so it is not possible to determine its impact on forecasting. But let's suppose the predictability, as measured by persistence, has increased by one day (which is probably optimistic) in the largest scales and the increase is spread over more than a century: in practice this will have unnoticeable impact for the Numerical Weather Prediction (NWP) challenge.

Especially considering that the mid-latitude intrinsic predictability limit could be attained by NWP systems well before the end of this century. Modern deterministic NWP systems have improved their skill for sea-level pressure forecast in average by one day for each decade since the 60's. We should expect future medium-range ensemble weather prediction systems to be close to their skill saturation by mid-century for synoptic scale.

In addition we expect with climate change an increase frequency of extreme large scale events and their impacts. That will make the life of forecasters in fact more difficult for the rest of the century. So in my view there is no evidence in this paper that weather forecasting is getting easier.

Minor comments:

- 1) P.6 : "positive and neutral" should be "neutral and positive"
- 2) P.20, Caption: We should read "(a,c,e).

Dear Editor,

We have revised the article now entitled “The hammam effect: how a warm ocean enhances large scale atmospheric predictability”, taking into account the observations raised by the reviewers. A point-by-point answer is provided below. We have also rewritten the abstract to respect the length limit and the format of Nature Communications and adjusted the references in the main text. We hope that the paper is now suitable for publication in Nature Communications.

Sincerely,

Davide Faranda

(on behalf of the authors)

Reviewer #1 (Remarks to the Author):

The hammam effect: how a warm ocean makes weather forecasting easier
By D. Faranda, M.C. Alvarez-Castro, G. Messori, D. Rodrigues and P. Yiou

General comments

I recommend that this paper be published, if my main concerns are properly addressed.

We thank the referee for the positive comments. We have taken into account all the issues raised and have modified the manuscript accordingly.

This study addresses the evolution of North Atlantic weather predictability within the framework of climate change. It claims using centennial reanalysis datasets and CMIP5 simulations (1851-2100 period) that weather predictability for the North Atlantic region should increase. The main explanation provided for this positive trend is warmer North Atlantic SSTs that would make more zonal the atmospheric circulation hence more predictable. The methodology of the diagnostics is relatively novel and promising (e.g., Faranda et al., Scientific reports, 2017). Overall I agree that the phase-space framework combined with proper statistical analyses is a capable approach to tackle this complex problem of predictability and dynamical processes.

I have no objection with the statistical methodology used in this work. My main concerns are the dynamical and physical interpretations of the findings and some unclear remarks. 1) What are the explicit values in 2000 for inverse persistence and local dimension in Table 1 and Figure 1? It is very difficult to assess the importance of these results with not having the absolute numbers. As an example in Figure 1a we see that the trends span at most one degree of freedom, but relatively to what?

For 20CR reanalysis and CMIP5 models, the absolute values are reported in [Rodrigues et al. 2018, JCLIM]. They all have the same order of magnitude as the NCEP reanalysis. The ensemble mean of the dimension varies between 12.0 and 12.4 among all the datasets. To provide the readers with an immediate reference, we have now included new Fig. S1 to the supplemental material, which displays the absolute

values. The decreasing trend of the local dimension we discuss in the study represents 5-8% of the absolute value. This is now explicitly discussed in the paper (lines 135-136) Please also refer to our answer below concerning the dynamical significance of the shift (*).

In Faranda et al. (Scientific reports, 2017) they provide a local dimension of around 12-13. This is consistent with the results of Brunet (JAS, 1994) that demonstrate that the low frequency variability (subseasonal to seasonal, S2S, time scale) is spanned by around 8 empirical normal modes in the North Hemisphere with growth rate ranging 3-5 days in phase space. The proposed methodology in this study is not able to capture the small-scale phenomena as mention rightly in Faranda et al. (Scientific reports, 2017), but the numerical weather prediction problem lies in great part in that variability (e.g., transient storms). In Brunet (JAS, 1994) small-scale phenomena project mostly into the phase speed continuous spectrum which is associated with transient atmospheric phenomena. So this study is about S2S predictability (e.g., weather regimes), not about weather predictability.

The Reviewer raises an important point, and we thank him/her for bringing the paper by Brunet, [1994] to our attention. Indeed our study relates primarily to the predictability of large-scale motions. To reflect this, we have substituted mentions of “change in weather predictability” with “changes in predictability of large-scale atmospheric motions” throughout the paper.

However, we note that our approach does also have a relation to conventional weather forecasts from meteorological to medium-range timescales, as also mentioned by the reviewer in his point (5). In Faranda et al. [2017 Sci Rep] and in Scher and Messori [2018 QJRMS], we have seen that there is a correlation between the variations of the dynamical indicators and the spread in forecasts of SLP fields for numerical weather prediction models. The information on the forecast error in the dynamical indicators is comparable (in order of magnitude) to that extracted with much (computationally) costlier machine learning techniques [Scher and Messori, 2018 QJRMS]. We have now commented this in lines 213-215.

In addition if we assume that the S2S component of the local attractor as a dimension of the order $O(10)$ as discussed in Faranda et al. (Scientific reports, 2017) and Brunet (1994) then we need to explain convincingly here why a small reduction of the dimension of the local attractor by less than one degree of freedom seems to have relatively an important impact on S2S predictability and the circulation configuration (e.g., more flow zonality). Are these two results simply fortuitous or wrong?

(What follows here also answers the Reviewer’s previous comment marked by *) The Reviewer correctly questions the significance of the shift found. We believe that the results are not fortuitous and we have some strong evidence to this effect given by a study of jet stream dynamics [Faranda et al. 2018, ESSD] where jet position data from ERA-Interim are embedded to derive a coupled lattice model of the jet dynamics. In that study, we found that the jet dynamics is very sensitive to small shifts in the control parameter (the meridional temperature gradient) and that the dimension tracks these changes, being proportional to the number of discontinuities of the jet (a proxy of blocking). Furthermore, we have found (Figure 9 in [Faranda et al. 2018, ESSD]) that the observed jet dynamics correspond to a region of the parameter space of our model that is very sensitive to perturbations, where small changes in the dimension

correspond to large dynamical changes. This is now discussed in the paper in lines 149-156.

2) The study at many occasions tries to explain the discrepancy of results between different reanalyses and modelling systems. The discussion is not convincing. Are these reanalyses and numerical simulations up to the task to detect such trends in the dimension of the local attractor and in the persistence? In Shepherd (Nature Geoscience 7, 2017) the case is clearly made that dynamical processes, like regional circulation over the North Atlantic, are prone to significant inconsistency between different climate models due to model error, especially with precipitation processes.

The reviewer is right about the misrepresentation of the precipitation processes. However, we found in Rodrigues et al. [2018, JCLIM] that models and the 20CR reanalysis provide a very consistent representation of the SLP fields, at least for the historical period. We believe that rephrasing the results in terms of large-scale predictability helps avoiding this confusion.

3) I don't understand the purpose of this discussion about SST local dimension. Why an increase in SST local dimension should be consistent with a decrease in the atmosphere local dimension?

In the new version of the manuscript, we have rephrased this part (lines 171-175). The direction of the change is not important here. What really matters is that there is a shift in SST dimension for +4K simulation and not for 4xCO2 simulations, consistently with what is observed for SLP.

4) P. 9 "The warmer SSTs also induce more predictable boundary conditions for atmospheric motions." Why?

We have rephrased this sentence as "Warmer SSTs also correspond to more predictable atmospheric configurations, since d_{SLP} is lower for warmer ocean states" (lines 190-192).

5) The title is misleading. As mentioned above this study is not about weather prediction (i.e. small scale phenomena like storms), but prediction of the large scale atmospheric features of the North Atlantic (e.g., blockings) which of course indirectly modulate in part the weather predictability. In addition the study doesn't quantify the predictability trend in absolute numbers, so it is not possible to determine its impact on forecasting. But let's suppose the predictability, as measured by persistence, has increased by one day (which is probably optimistic) in the largest scales and the increase is spread over more than a century: in practice this will have unnoticeable impact for the Numerical Weather Prediction (NWP) challenge.

Especially considering that the mid-latitude intrinsic predictability limit could be attained by NWP systems well before the end of this century. Modern deterministic NWP systems have improved their skill for sea-level pressure forecast in average by one day for each decade

since the 60's. We should expect future medium-range ensemble weather prediction systems to be close to their skill saturation by mid-century for synoptic scale.

In addition we expect with climate change an increase frequency of extreme large scale events and their impacts. That will make the life of forecasters in fact more difficult for the rest of the century. So in my view there is no evidence in this paper that weather forecasting is getting easier.

We have modified the title to reflect the fact that we mostly investigate large-scale predictability. The new title is “The Hamman effect: how a warm ocean enhances large scale atmospheric predictability”. However, as specified in our above answer, there is indeed a relatively close relation between the dynamical indicators and weather predictability on meteorological timescales. We have added a discussion about this in lines 213-215.

We further note that, if indeed ensemble weather prediction systems will reach close to their skill saturation by mid-century, future changes in intrinsic predictability will be all the more important, as they won't occur on the background of rapid and constant improvements to the models themselves.

Minor comments:

- 1) P.6: “positive and neutral” should be “neutral and positive”
- 2) P.20, Caption: We should read “(a,c,e).

We thank the referee for spotting these typos, which we have now corrected.

Reviewer #2 (Remarks to the Author):

The hammam effect: how a warm ocean makes weather forecasting easier
by Davide Faranda et al.. Submitted to Nature Communication

This manuscript describes results obtained through a technique developed to evaluate the predictability of the weather system under increased anthropogenic forcing associated with climate change and global warming. This is done through re-analysis of three historical data sets, and climate simulations. In other words, the paper asks how large temporal scales can affect small weather scales. This is performed through an evaluation of the number of active degrees of freedom of the atmospheric attractor, through two indices: dimension and (inverse) persistence. In the supplemental part of the paper, an analysis is also performed by season. This analysis is then pursued on future events for forced simulations with 4 times CO₂ or four times warmer ocean, the latter being significantly correlated with the results whereas enhanced greenhouse does not seem to be relevant in this context.

The introduction is very clearly written and makes you want to know in more detail what has been accomplished. I find the result very interesting: there is likely an increase in the predictability of the atmosphere, and this is attributed to warmer oceans as a driving force.

But if I were to associate a warmer climate thus a stronger forcing with a higher dimensionless governing parameter, such as a Reynolds number based on the forcing amplitude, then I expect stronger chaos, certainly in the small scales and thus less predictability, again at least in the small scales. How can one solve this (apparent) contradiction? And, perhaps, is this increased predictability of the large-scale flow associated or not with a decreased predictability of the small scales? Would that correlate with stronger hurricanes for example? What seems to be behind the mechanism is that, in my terms as a physicist, the surplus energy due to global warming is fed to the large scales of the oceanic flow (and likely to the large-scale atmospheric flow as well, as hinted at the end of the paper) because of rotation; large scales have more predictability in particular if rotation is an agent in this system leading to a non-intermittent inverse energy cascade at high latitudes. Of course, the tropics might be another story.

The reviewer raises an important point and we have thought about it for a long time. When we began this study, we speculated that, for exactly the same reasons explained by the Reviewer, we would have observed an increase in the dimension. This would have been consistent with the picture proposed by the Reviewer. However, there are systems where an increase of the Reynolds number leads to more predictable states. A first example is the von Karman flow dynamics: in Faranda et al. [2017, PRL], we have shown that a dynamics with a small number of degrees of freedom emerges for very high Reynolds numbers. In particular, we go from a very high dimensional, noisy fixed point to a stochastic chaotic attractor, where the dimension is smaller and the large scale dynamics can be described by Duffing equations. Another example is provided in Faranda et al. [2018, ESSD]. There, we analyse the jet stream dynamics (which is closely related to this study), using a low dimensional model obtained by embedding ERA-Interim reanalysis data. The bifurcation sequence, when increasing

the Reynolds number, is similar to that observed in the von Karman experiment: we go from a high dimensional noisy fixed point to a lower dimensional structure where the dynamics of the jet consist of switching between blocked and zonal flows. More generally, there is a new and interesting field that tries to explain these phenomena (see [Matsumoto et al J Phys Stat 1983]) and it is known as “noise induced order”. We have now clarified this important point in the paper, i.e. that increasing the Reynolds number can induce more predictable motions, which is of interest also for colleagues working in turbulence. These changes appear in lines 193-204.

Some rather minor questions and remarks:

- In the abstract, “most datasets point to an increase in the atmospheres intrinsic predictability.” Do the authors mean at the weather or at the climate scales?

The sentence is ambiguous.

We agree that we should be more specific. We look at predictability at weather scales, because the local dimensions and persistence are measured for sea-level pressure fields on a daily basis. Then we look at the long-term trends of this predictability. We have rephrased the abstract as “We evaluate the changes in large-scale atmospheric circulation predictability in the sub-seasonal range over the North-Atlantic under anthropogenic forcing using centennial reanalyses and CMIP5 simulations over 1851-2100.”

- I find there is something wrong with the following sentence: “Both are local in phase-space and instantaneous in time and issue from a recently developed extreme value theory ...”

We have rephrased the sentence to “The two dynamical systems metrics we use are the local (in phase space) dimension d and the persistence θ^{-1} (20: Lucarini et al 2016)”. Lines 74-75

- Figure 1a, for d , why is 20-CR so different, at least at early times, and why does it have larger fluctuations (also for θ data)? And CERA20C?

Before the 1950s’, the 20CR data are constrained by a smaller number of observations, we have extensively commented this issue in Rodrigues et al. [2018, JCLIM]. The reanalysis datasets experience larger fluctuations than the other datasets, because they are constrained by observations. Therefore, even if there are several ensemble members, they are forced to follow the same dynamics, whereas the internal variability on decadal time scales in the models may be similar to those of the reanalysis, but it is averaged out within the ensemble. We have added this explanation. See lines 118-121.

- What is zonality? I suppose more zonal large-scale quasi-2D flow like a jet?

Indeed, we have added this definition as: “By large-scale zonal flow we refer to a strong eastward (zonal)jet with few meridional oscillations”. Lines 36-37

- Figure 3, it is quite striking indeed that high inverse persistence and high dimension comes with lower SSTs, with a difference of at least one Kelvin.

Indeed!

- Tables 1, S1 and S2: A brief description of the Mann-Kendall test p-values would be welcome. It appears nowhere else in the paper as far as I can tell.

We have added this definition in the supplementary material

- Definitions of local dimension d and persistence θ :

In the Method section, I have difficulties with the computation of the dimension d : (i) what is the link with the old-fashioned way of computing dimensions of attractors; (ii) I find that in equations (1,2), x as well as σ , μ are undefined; (iii) where does θ appear?

We have rephrased this section to better explain the parameters and to account for the relation between the computation of the local dimensions and the “old-fashioned” computation by, e.g. Liebovitch et al [Phys Lett 1989]

- Similarly, in the Persistence paragraph in the Methods Section, define “immediately”: is it at the next Δt ? How does that depend on the choice of temporal discretization, then?

Yes, it depends on the discretization - this is an important point that we now explicitly mention.

- A detail: θ is defined as a time, you just cannot introduce it in an exponential; then what is the normalization time for the process? How does it depend on the discretization time, or rather does their ratio matter in any way in computing these indicators? How robust is the method?

The reviewer is right. We have expanded the explanation of the normalization, introducing the explicit dependence on Δt in the definition of θ , whereas in the exponential the adimensional extremal index ϑ now appears. The method is robust in the following sense: if we upgrade the time resolution by a factor 4, considering for e.g. 6-h sea-level pressure fields, then we obtain that the values of θ are all divided by 4. This also suggests that it is better to have a time sampling such that θ varies between 0 and 1. A time sampling with too large a Δt will cause $\theta=1$ always, and therefore there will be no persistence information in the dynamics.

- Trivial remarks such as, p. 7: idea, We
- Caption, Fig. 2: and and
- Caption, Fig. 3: (acE) and (bdf)
- Caption of Table 1: significant

In conclusion, this paper is interesting, with important potential consequences in assessing the behavior of the atmosphere such as, for example, atmospheric blocking. It is well written.

In my opinion, it should certainly be published after the above remarks are taken into account.

Thank you, we have corrected these typos.

Annick Pouquet, National Center for Atmospheric Research

1) What are the major claims of the paper? Atmospheric dynamics is more predictable under global climate warming. Technically, this increased predictability is linked to a lower attractor dimension, with more “zonality” (see above).

2) Are they novel and will they be of interest to others in the community and the wider field? Absolutely.

3) Is the work convincing? Yes, the data is excellent.

4) Will the paper influence thinking in the field? It certainly will, as attested by the wide array of journals in the references: human live at the small weather scale; different communities (meteorology, oceanography, climate, theoretical and statistical physics, data analysis) are a priori involved.

5) Appropriateness and validity of statistical analysis: authors are expert in the field.

6) Ability of a researcher to reproduce the work, given the level of detail provided: yes (but see question above).

We thank the Reviewer for her comments.

Reviewer #1 (Remarks to the Author):

The authors have provided convincing rationale. The paper is ready for publication.

Reviewer #2 (Remarks to the Author):

I am satisfied with the changes made to the manuscript and suggest that the paper be published.